

# One-pot multicomponent nitro-Mannich reaction using a heterogeneous catalyst under solvent-free conditions

Giovanna Bosica and Ramon Zammit

Department of Chemistry, University of Malta, Msida, Malta

## ABSTRACT

An environmentally-friendly, one-pot multicomponent reaction of various aldehydes, amines and nitroalkanes for the synthesis of β-nitroamines is here described. Amberlyst A-21 supported CuI was found to be a highly efficient novel heterogeneous catalyst for the three-component nitro-Mannich reaction between aldehydes, amines and nitroalkanes. The developed protocol is performed in a solvent-free medium to produce a variety of β-nitroamines in good to excellent yields within short reaction times. The catalyst can be easily prepared and recovered. It has been tested up to eight times with only a minor activity loss.

## INTRODUCTION

The synthesis of bioactive secondary metabolites through the formation of networks of carbon–carbon bonds has fascinated synthetic organic chemists for entire generations (*Kolb, Finn & Sharpless, 2001*). The nitro-Mannich reaction is a Mannich-type reaction (Fig. 1), an imine-based reaction, where the original nucleophile, an enolisable carbonyl compound, has been replaced by a nitroalkane.

It is a powerful C–C bond forming reaction which involves the addition of nucleophilic nitronate species, derived from nitroalkanes, to electrophilic imines (*Noble & Anderson, 2013*; *El-Sayed, Mahmoud & Hilgeroth, 2013*). The resulting β-nitroamine product is considered to be a privileged building block because it possesses potential vicinal stereocentres, bearing nitro and amino functional groups with different oxidation states, providing opportunities for the selective manipulation of either in subsequent transformations (*Anderson & Koovits, 2013*; *Marqués-López et al., 2009*). β-Nitroamines serve as useful intermediates to access other valuable moieties such as 1,2-diamines (via nitro reduction) (*Westermann, 2003*), α-amino carbonyls and α-amino acids (via the Nef reaction) (*Ballini & Petrini, 2004*), peptides synthesis of umpolung chemistry (*Shen, Makley & Johnston, 2010*) and monoamines (via reductive denitration). The nitro functional group can also form other functionalities such as hydroxylamines (*Ruano et al., 2006*), and oximes, or nitriles (*Czekelius & Carreira, 2005*). Indeed several natural products and pharmaceuticals have been synthesised via an intermediate nitro-Mannich reaction including antiemetic (*Tsuritani et al., 2002*), analgesics (*Kumaraswamy & Pitchaiah,*

Corresponding author
Giovanna Bosica,
giovanna.bosica@um.edu.mt

**Figure 1** Mannich reaction between aldehydes, amines and ketones.

*2011a*), antimalarial and antiinsecticidal (*Jakubec et al., 2012*), antipsychotic (*Handa et al., 2010*; *Davis, Danneman & Johnston, 2012*), antiviral (*Weng et al., 2010*), anticancer (*Davis & Johnston, 2011*; *Vara et al., 2014*), anti-HIV (*Xie et al., 2011*), antiparasitic (*Choudhary et al., 2014*), antidepressant (*Hynes, Stupple & Dixon, 2008*), antibacterial and antimicrobial (*Jakubec, Cockfield & Dixon, 2009*), and anaesthetic (*Kumaraswamy & Pitchaiah, 2011b*).

Yet despite the utility, research in the nitro-Mannich reaction has not been very popular because the β-nitroamine product is often prone to retro-addition and epimerisation (*Walvoord & Kozlowski, 2015*). Moreover, the addition of nitronates to imines is not thermodynamically favoured due to the difference between the p$K$a values of nitroalkane (p$K$a ∼9) and the β-nitroamine product (p$K$a ∼35) (*Noble & Anderson, 2013*). As a result, Lewis or Brønsted acids were suggested to activate the electrophilic imine (*Bernardi et al., 2003*; *Anderson et al., 2005*; *Pelagalli et al., 2016*), or organic and inorganic bases to activate the nitroalkane, to form the nitronate species (*Adams et al., 1998*; *Rodríguez-Solla et al., 2012*; *Wang et al., 2014*; *Kutovaya et al., 2015*). These catalysts, however, present a number of disadvantages since big amounts are applied, certain metals used are very expensive, and only function in harsh temperature conditions, with large excess of nitroalkane and in the presence of hazardous solvents. In this regard, there has been a strategic focus to develop more environmentally friendly nitro-Mannich protocols. This has been achieved in two ways: the employment of a heterogeneous or recyclable catalyst (*Ballini et al., 2014*; *Komura et al., 2010*; *Chakrapani & Kantam, 2011*), which can be recovered and reused, or the application of a one-pot multicomponent strategy between aldehydes or ketones, amines and nitroalkanes (Fig. 2) (*Soengas & Silva, 2013*; *Piscopo et al., 2013*; *Cruz-Acosta, De Armas & Garcia-Tellado, 2013*).

Heterogeneous catalysts have been shown to also catalyse like the homogeneous ones with some advantages. In fact they are non-corrosive, safely handled, porous material increasing the surface area, result in more effective encounter rates, are easily recovered from the reaction mixture without the need for reaction quenching and can be recycled and reused (*Sharma & Singh, 2011*). Following our studies for the development and application of heterogeneous catalysts in multicomponent reactions (*Bosica & Abdilla, 2017b*), herein we wish to report a simple and efficient approach to the one-pot three-component nitro-Mannich reaction using a recyclable heterogeneous catalyst.

**Figure 2** One-pot multicomponent nitro-Mannich reaction between aldehydes or ketones 1, amines 2 and nitroalkanes 3.

| Entry[a] | Catalyst | Time (h) | Yield 6 (%)[b] | Entry[a] | Catalyst | Time (h) | Yield 6 (%)[b] |
|---|---|---|---|---|---|---|---|
| 1 | No | 168 | -[c] | 14 | KF-silica (53 wt%) | 168 | 39 |
| 2 | K$_2$CO$_3$ (20 mol%) | 120 | 49 | 15 | KF-montmorillonite (14 wt%) | 168 | 51 |
| 3 | Na$_2$CO$_3$ (10 mol%) | 96 | 50 | 16 | KF/ basic alumina (33 wt%) | 120 | 54 |
| 4 | Cs$_2$CO$_3$ (200 mol%) | 110 | 10 | 17 | CuI/Amberlyst A-21 (5 mol%) | 48 | 75 |
| 5 | NaHCO$_3$ (10 mol%) | 120 | 50 | 18 | CuCl/A-21 (5 mol%) | 96 | 35 |
| 6 | KOH (100 mol%) | 120 | 49 | 19 | AgNO$_3$/A-21 (5 mol%) | 168 | 18 |
| 7 | Silica (332 mol%) | 48 | 69 | 20 | CuI/Montmorillonite K-10 (5 wt%) | 168 | 21 |
| 8 | Basic Alumina (400 mol%) | 72 | 59 | 21 | CuI/basic Alumina (3 mol%) | 40 | 50 |
| 9 | Amberlyst A-21 (33mg/ 1mmol of aldimine) | 96 | 39 | 22 | CuI-doped basic Alumina (200 mol% CuI and 400 mol% alumina) | 40 | 22 |
| 10 | Amberlite® IRA-400 Cl⁻ (0.5g/1.5 mmol of Cl⁻) | 168 | 50 | 23 | CuI (5 mol%) | 120 | 23 |
| 11 | Nafion SAC-13 (30 wt%) | 120 | 32 | 24 | Ag(I)/Montmorillonite-K-10 (7 wt%) | 168 | 36 |
| 12 | Montmorillonite K-10 (20 wt%) | 96 | 45 | 25 | PEG-400 (5 mL) | 120 | 29 |
| 13 | PPA/SiO$_2$ (20 wt%) | 72 | -[c] | | | | |

**Figure 3** Catalyst screening tests carried out on the nitro-Mannich step for the synthesis of 6 using previously prepared *N*-benzylideneaniline (5) and nitromethane (3a). The highlighted row in green indicates the preferred catalyst used for subsequent optimisation tests. a, All reactions were carried out at room temperature, in a 1:10 5:3a ratio at 2.5 mmol. b, Isolated yield. c, No product.

## RESULTS AND DISCUSSION

### Catalyst screening

Initial investigation involved employing a series of catalysts to screen the nitro-Mannich step between *N*-benzylideneaniline, (5, 1 mmol) and nitromethane, (3a, 10 mmol) as model reaction (Fig. 3). Due to the relative ease in the formation of the intermediate imine, screening for catalyst selection was only performed on the second more challenging nitro-Mannich step, from preformed imines. Various heterogeneous catalysts or catalytic species which could be immobilised on a solid support were all employed under solvent-free conditions. Both organic and inorganic bases, and Lewis or Brønsted acids were used to potentially activate the nucleophilic nitroalkane or electrophilic imine, respectively.

**Figure 4** One-pot sequential multicomponent nitro-Mannich reaction between benzaldehyde 1a, aniline 2a, and nitromethane 3a to form β-nitroamine product 6.

Catalysts tested include species which have been previously tried in the nitro-Mannich reaction such as carbonates, bicarbonate and hydroxide inorganic salts (entries 2–6) (*Wang et al., 2008*; *Zhang, Liu & Liu, 2011*; *García-Muñoz et al., 2014*), silica (entry 7) (*Mahasneh, 2006*), basic alumina (entry 8) (*Baricordi et al., 2004*), and KF on alumina (entry 16) (*Ballini et al., 2014*), as well as species which have been documented to function efficiently in similar Mannich-type reactions such as Amberlite IRA-400 Cl resin (entry 10) used to catalyse the aza-Friedel Crafts reaction (*Harichandran, Amalraj & Shanmugam, 2016*), montmorillonite K-10 (entry 12) used to synthesise numerous Mannich bases (*Arunkumar, Subramani & Ravichandran, 2010*), PPA-silica (entry 13) used in double Mannich condensation reactions (*Vekariya, Prajapati & Patel, 2016*) or similar C-C bond forming reactions such as Amberlyst A-21 (entry 9) used to catalyse the nitroaldol reaction (*Ballini, Bosica & Forconi, 1996*). Since modest results were observed with different acidic and basic strengths, a number of bifunctional catalysts were tested. Amberlyst A-21 supported copper(I) iodide (entry 17) was previously used in the $A^3$- and $KA^2$-coupling reactions, an acetylene-Mannich reaction (*Bosica & Gabarretta, 2015*; *Bosica & Abdilla, 2017a*). Given the Lewis acid $Cu^+$ ions and basic tertiary amino-grafted polystyrene polymer, it was postulated that its activity could also be extended to the formation of β-nitroamines. Complete conversion for the synthesis of **4a** occurred at a faster rate (48 h) and the product was isolated with a good yield, confirming the hypothesis. Substitution of the transition metal, (entry 19), metal counter ion, (entry 18), or metal support, (entries 20, 21), resulted in a reduction of the catalytic activity or lower product yield. As a result, Amberlyst A-21 supported CuI was selected for the basis of this study.

## Reaction optimisation

In order to develop a greener approach to the nitro-Mannich reaction, efforts were made to employ Amberlyst A-21 supported CuI in a one-pot three-component strategy reaction between benzaldehyde (**1a**) and aniline (**2a**) (the starting materials required to form *N*-benzylideneaniline, **5**) together with nitromethane (**3a**) (Fig. 4). The multicomponent reaction was performed in a sequential manner such that the catalyst, CuI-Amberlyst A-21, **1a** and **2a** were first mixed to form the intermediate **5** followed by addition of **3a** in the same pot to form the resultant β-nitroamine **6**. Using the same reaction conditions as the two-step strategy, the reaction was complete after 48 h and the isolated yield of **6** was slightly improved, confirming the efficiency of the heterogeneous catalyst to catalyse the multicomponent reaction.

| Entry | Aldehyde[a] 1a (mmol) | Amine[a] 2a (mmol) | Nitroalkane[a] 3a (mmol) | Catalyst (mol%) | Temperature (°C) | Time (h) | Yield[b] 6 (%) |
|---|---|---|---|---|---|---|---|
| 1[c,d] | 1 | 1 | 10 | 5 | RT | 48 | 78 |
| 2[d] | 1 | 1 | 10 | 20 | RT | 48 | 74 |
| 3[d] | 1 | 1 | 10 | 10 | RT | 48 | 72 |
| 4[d] | 1 | 1 | 10 | 2 | RT | 60 | 44 |
| 5[d] | 1 | 1 | 10 | 1 | RT | 60 | 43 |
| 6[d] | 1 | 1 | 7.5 | 5 | RT | 48 | 74 |
| 7[d] | 1 | 1 | 5 | 5 | RT | 48 | 82 |
| 8[d] | 1 | 1 | 2.5 | 5 | RT | 48 | 62 |
| 9[d] | 1 | 1 | 1 | 5 | RT | 48 | 40 |
| 10[d] | 1 | 1 | 5 | 5 | RT under $N_2$ | 60 | 67 |
| 11[e] | 1 | 1 | 5 | 5 | 100 | 2 | 86 |
| 12[e] | 1 | 1 | 5 | 5 | 80 | 5 | 84 |
| 13[e] | 1 | 1 | 5 | 5 | 60 | 10 | 80 |
| 14[e] | 1 | 1 | 5 | 5 | RT | 48 | 81 |
| 15[e] | 1 | 1 | 4 | 5 | 100 | 2 | 70 |
| 16[e] | 1 | 1 | 3 | 5 | 100 | 2 | 62 |

**Figure 5** **Optimisation trials on the model nitro-Mannich reaction between benzaldehyde (1a), aniline (2a) and nitromethane (3a). Highlighted cells indicate a change in factor. Cell highlighted in red indicates the entry with ideal conditions.** a, Reaction was performed on a 2.5 mmol scale. b, Isolated product yield. c, Original conditions in which copper (I) iodide was used during the screening of catalysts on the model reaction. d, Trial was done using a batch of catalyst which was previously prepared and stored in a desiccator under vacuum with a loading of 1.35 mmol CuI g$^{-1}$. e, Trials were done using a newly prepared batch of catalyst with a loading of 1.52 mmol CuI g$^{-1}$

Efforts were then made to optimise the reaction conditions of the model multicomponent reaction. A number of factors were evaluated including catalyst loading, temperature, and molar ratio, in particular the amount of excess nitroalkane (Fig. 5). Using as little amount of catalyst as possible is essential to minimise its cost. In fact, no improvement in activity or yield was observed when increasing the catalyst loading to 10 and 20 mol% (entries 2, 3) from the original 5 mol% used (entry 1). However, further decreasing the catalyst amount to 2 and 1 mol% (entries 4, 5) resulted in a negative effect both on the reaction time and yield. Concerning the molar ratio, the reaction is preferably performed in a stoichiometric ratio of starting material since excess reagents increase the E-factor, resulting in a less green process. Whilst maintaining the aldehyde and amine to a 1:1 stoichiometric ratio, excess nitroalkane was necessary since its deprotonation to form the stable nitronate anion is slow and thus an excess of nitroalkane is used to increase the rate of deprotonation. A compromise was found by performing optimisation trials to reduce the amount of

| Entry[a] | Solvent[b] | Temperature (°C) | Time (h) | Yield 6 (%) |
|----------|-----------|------------------|----------|-------------|
| 1 | - | 100 | 2 | 86 |
| 2 | toluene | 100 | 2 | 28 |
| 3 | chloroform | reflux | 2 | 20 |
| 4 | ethyl acetate | reflux | 2 | 25 |
| 5 | acetonitrile | reflux | 2 | 28 |
| 6 | methanol | reflux | 2 | 84 |
| 7 | water | 100 | 2 | 79 |

**Figure 6 Effect of solvent on the multicomponent nitro-Mannich reaction between benzaldehyde (1a), aniline (2a) and nitromethane (3a).** a, Reaction was performed at 2.5 mmol scale with a 5-fold excess 3a. b, 5 mL solvent was added in each trial.

excess nitromethane used. Reducing the original 10-fold excess of 3a used up to 5-fold excess resulted in the same conversion rate and a slightly improved yield of 82% (entry 7). However, further reducing the amount of excess 3a (entries 8, 9) resulted in a deleterious effect on both the reaction time and yield. Performing the reaction under an atmosphere of nitrogen resulted in a slower reaction time, with complete conversion occurring only after 60 h and the isolated yield also decreased (entry 10). This was important to determine that the reaction is insensitive to oxygen, thus ensuring simple conditions. Since the catalytic activity was not highly efficient at room temperature, the reaction was heated at different temperatures to increase the rate of reaction. Heating the reaction at 100 °C resulted in completion after 2 h and a very high isolated yield was obtained (entry 11). Lower temperatures, 80 and 60 °C, resulted in longer reaction times and lower yields (entries 12, 13).

Since a new batch of catalyst was used half way through the optimisation trials, a comparison between the two batches was performed by repeating the reaction at room temperature with the new batch (entry 14). This resulted in similar reaction times and yields indicating that increased copper (I) iodide loading did not have any negative consequences on the activity at room temperature.

The effect of solvent on the multicomponent nitro-Mannich reaction was next examined. Solvents from the three major classes: non-polar, polar aprotic and polar protic solvents were used. Non-polar solvents (Fig. 6, entries 2, 3) and polar non-protic solvents (entries 4, 5) resulted in a poor conversion whilst polar protic solvents (entries 6, 7) exhibited good catalytic efficiencies, with high product yields. Despite this, there was no improvement in yield compared to the solvent less reaction and thus the latter more environmentally-friendly condition was maintained.

## Examination of substrate scope: nitro-Mannich Reaction with different aldehydes, amines and nitroalkanes

The generality of the one-pot nitro-Mannich reaction was explored using a 5 mol% CuI on Amberlyst A-21 catalyst (with a loading of 1.52 mmol CuI g$^{-1}$), a 5-fold excess of nitroalkane, heating at 100 °C in the presence of air and using solvent-free conditions. A series of aldehydes, amines and nitroalkanes were subjected to the optimised conditions.

Pleasingly, the reaction was shown to tolerate a wide variety of aromatic aldehydes with substituents on the main ring in moderate to excellent yields (Fig. 7, entries 1–10), whilst aliphatic aldehydes were less reactive and resulted in poor to good yields (entries 11–13). Overall, the reaction times for the successful trials performed at 100 °C were short, irrespective of whether aromatic or aliphatic aldehydes were used. The electronic property of the substituent has a certain effect on the yield. In fact, aromatic aldehyde **1f** (entry 5), with a strong electron-donating group, resulted in a moderate yield since this results in weak aldehyde and imine electrophiles. It is interesting to note that a moderate electron-donating methoxy group in the *para* position (entries 2, 10) resulted in very good to excellent isolated product yields. Despite lowering the electrophilicity of the aldehydes and imines, the methoxy group facilitates complexation of the Lewis acid Cu$^+$ catalyst ions, activating the electrophilic aldehyde for nucleophilic attack by the aniline and the intermediate electrophilic imine for attack by nitromethane (*Dalpozzo et al., 2006*). The fact that the strong electron-donating group (entry 14) did not form the nitro-Mannich product indicates that there is a balance between the opposite forces exerted by the electrophilicity of the carbon and the coordination of the Cu$^+$ ions to the oxygen and nitrogen atoms in the C=O and C=N bonds, respectively. Electron-withdrawing halogen-substituted benzaldehydes (entries 3, 8, 9) resulted in moderate to good yields, overall lower than the model reaction. Despite making the aldehyde and imine more electrophilic, they slow the activation mechanism of the catalyst. Aliphatic aldehydes resulted into poor yields due to being stronger electron-donating groups and also tend to make the intermediate aliphatic imine more unstable by tautomerisation to the enamine, and hence undergo side reactions. The scope of the reaction was also tested with two ketones (entries 15, 16); however, no product was collected due to lower electrophilicities and additional steric hindrance.

Nitro-Mannich products with variation in the aromatic amine substituent (Fig. 8) resulted overall in short reaction times, with moderate to excellent yields. Both electron-donating **2b-f** and electron-withdrawing groups **2g-l** were good substrates and fast reaction times and good yields were achieved. Again, such results can be explained in terms of the balance between the increased electron density on the C=N bond of the intermediate caused by electron-donating groups, which increased the likelihood of activation by complexation with the Lewis acid Cu$^+$ catalyst despite reducing the electrophilicity, and the increase in the electrophilic strength of the imines with electron-withdrawing groups. Aromatic amines **2f** and **2l** resulted in lower yields due to increased steric hindrance. Aliphatic amines (entries 13, 14) resulted in no product formation due to the less stable imines and β-nitroamine products.

A set of nitro-Mannich reactions (Fig. 9) focused on investigating other nitroalkanes. Increasing the length of the alkyl moiety of the nitroalkane using **3b** and **3c** resulted in

**Figure 7  Substrate scope with various aldehydes for the synthesis of β-nitroamines 7a-p.**  a, isolated pure yield; b, No product formation.

**Figure 8** **Substrate scope with various amines for synthesis of β-nitroamines 8a-n.** a, isolated pure yield; b, no product formation.

**Figure 9 Substrate scope with various nitroalkanes for the synthesis of β-nitroamines 9a-b.** a, isolated pure yield; b, determined by $^1$H NMR spectroscopy or from the isolated yield when separation between the two pairs of diastereomers was possible.

an increase in reaction time and a decrease in the isolated yield due to a decrease in the stability of the nitroalkane carbanion. Both reactions afforded a good diastereoselectivity (4:1).

To further shed light on the versatility of the multicomponent nitro-Mannich reaction, substitution of more than one substrate at a time was performed. Overall, the reaction time for the successful trials performed at 100 °C was fast, between 1.5–4 h, and moderate to excellent yields were obtained in most of trials (Fig. 10). Products **4b**, **4e** and **4i** (entries 2, 5, 9) resulted in short reaction times and high yield, longer reaction times were observed for products **4g**, **4j** and **4k** (entries 7, 10, 11) due to steric hindrance by the bulkier *p*-ethyl moiety and a longer nitroalkane chain. Aliphatic aldehydes **1m** and **1l** resulted in a diminished yield (entries 4, 8). No product was formed due to the bulky *ortho* and *meta* positioned nitro moieties in **4f** (entry 6) and 2,6-dichloro-substituted benzaldehyde **1j** (entry 12). Diastereoselectivity was again observed for products **4j** and **4k**, although to a lesser extent.

## Mechanistic aspects

A tentative catalytic cycle is proposed (Fig. 11). The reaction consists of two crucial steps, the condensation step involving imine formation and subsequent nucleophilic addition by a nitroalkane to the *in situ* formed imine to synthesise the final nitro-Mannich product. In the first step, the Lewis acid metal salt CuI, likely catalyses the nucleophilic addition of amines to aldehydes to form the intermediate hemiaminal. The Lewis acid also likely functions as a dehydrating agent to remove a water molecule, which is eliminated from the hemiaminal (*Dalpozzo et al., 2006*). This results in formation of the imine. The second

| Entry | Aldehyde | Amine | Nitroalkane | Product | Time (h) | Yield 4 (%)[a] | dr ratio[c] |
|---|---|---|---|---|---|---|---|
| 1 | 1b | 2k | 3a | 4a | 2 | 75 | -[d] |
| 2 | 1c | 2h | 3a | 4b | 1.5 | 94 | -[d] |
| 3 | 1f | 2b | 3a | 4c | 2 | 56 | -[d] |
| 4 | 1m | 2h | 3a | 4d | 2 | 67 | -[d] |
| 5 | 1e | 2c | 3a | 4e | 2 | 77 | -[d] |
| 6 | 1e | 2k | 3a | 4f | 12 | -[b] | -[d] |
| 7 | 1d | 2f | 3a | 4g | 3 | 67 | -[d] |
| 8 | 1l | 2c | 3a | 4h | 2 | 45 | -[d] |
| 9 | 1b | 2m | 3a | 4i | 2 | 84 | -[d] |
| 10 | 1c | 2i | 3b | 4j | 3 | 81 | 53:47 |
| 11 | 1e | 2c | 3c | 4k | 4 | 54 | 60:40 |
| 12 | 1j | 2d | 3c | 4l | 8 | -[b] | -[d] |

**Figure 10  Substrate scope with mixed starting materials for the synthesis of β-nitroamines 4a-l.**  a, isolated pure yield; b, no product formation; c, determined by $^1$H NMR spectroscopy; d, not available.

**Figure 11** A possible mechanism for the synthesis of β-nitroamine 12 by means of CuI.A-21 catalyst.

step involves the coordination of CuI to the imino nitrogen, thus activating the imine by formation of a positively charged nitrogen atom, making it a stronger electrophile. This type of activation is typical of copper catalysts used in the nitro-Mannich reaction (*Das et al., 2014*). Moreover, validity of the possible mechanism was shown by the fast reaction times and high yields obtained with electron-donating groups since these resulted in a greater electron density on the C=N, which helped the activation of the imine by coordinating with the Lewis acid Cu$^+$ catalyst. It was suggested that the CuI salt coordinates also to the nitroalkane oxygen, increasing its acidity and thus favouring deprotonation to form the reactive nitronate anion (*Das et al., 2014*).

The base deprotonating the nitroalkane was likely to be the Amberlyst A-21 support, since it contains a tertiary amine, which is a much stronger base than iodide anions in CuI. Hence, unchelated dimethylamino sites likely deprotonate the nitroalkane to form the reactive nucleophilic nitronate species, which attacks the activated imine to form the β-nitroamine product, and at the same time releasing the catalyst to be used for a next cycle.

### Environmental acceptability of the nitro-Mannich reaction

The greenness of the nitro-Mannich reaction was determined using two main parameters: the E-Factor and the atom economy. Being a multicomponent reaction, involving only catalytic amounts of catalyst, and consisting of a condensation and an addition step, which only generates water as a by-product, the developed protocol displays both a very high Atom Economy as well as a low E-Factor (based on the model reaction for the synthesis

|  | **Unsupported CuI** | **CuI.A-21** |
|---|---|---|
| **Reagent ratio (mmol)[a]** | 2.5 (aldehyde) : 2.5 (amine) : 12.5 (nitroalkane) | |
| **Catalyst Quantity** | 5 mol% | |
| **Yield (%)[b]** | 28 | 86 |
| **Workup** | Direct Loading | Catalyst Filtration |
| **Waste** | 0.125 mmol CuI catalyst 10 mmol excess nitroalkane 2.5 mmol $H_2O$ waste | 10 mmol excess nitroalkane 2.5 mmol $H_2O$ waste |
| **E-factor** | 4.02 (including catalyst loss) | 1.26 (with complete catalyst recovery) |

**Figure 12** Comparison of the E-factor of unsupported CuI with Amberlyst supported copper(I) iodide for the synthesis of **6**. a, Based on the model reaction for the synthesis of **6** from **1a**, **2a** and **3a**. b, isolated yield.

of **6**, Eqs. (1) and (2)).

$$E\text{-Factor}_{\text{CuI.A}-21} = \frac{\text{Total Waste(g)}}{\text{Total Product(g)}} = \frac{0.610 + 0.045}{0.520} = 1.26 \qquad (1)$$

**Equation 1**: Low E-factor for the multicomponent nitro-Mannich reaction.

$$\text{Atom Economy} = \frac{\text{RMM}_{\text{product}}}{\sum \text{RMM}_{\text{reagents}}} \times 100 = \frac{242}{(106 + 93 + 61)} \times 100 = 93\% \qquad (2)$$

(RMM = Relative Molecular Mass)

**Equation 2**: High Atom Economy since only water is formed as by-product.

The effectiveness of the heterogeneous catalyst was demonstrated by comparison of the amount of waste generated with unsupported copper(I) iodide (Fig. 12). The results clearly indicate the advantage of using the polymer-supported heterogeneous catalyst, which not only resulted in a much higher isolated yield for **6** but also in a lower E-Factor, indicating that the reaction performed using a recoverable heterogeneous catalysis was more environmentally-friendly, in terms of the waste produced.

Amberlyst A-21 supported copper(I) iodide displays a number of advantages (Fig. 13) which makes it an interesting and effective heterogeneous catalyst in synthetic organic chemistry.

It is a cheap catalyst since both the metal salt and the dimethylaminomethyl-grafted polymer are commercially available. Preparation of the supported catalyst is simple and involves the drying of Amberlyst A-21 followed by chelation of CuI on the polymeric support (Girard et al., 2006). The chelation process takes approximately one day. The nano-sized copper salt so obtained, being immobilised on the terminal nitrogen in the dimethylamino moiety within the ion-exchange polymeric resin, results in an improvement in activity since the surface area for catalytic activity is increased and such chelation also

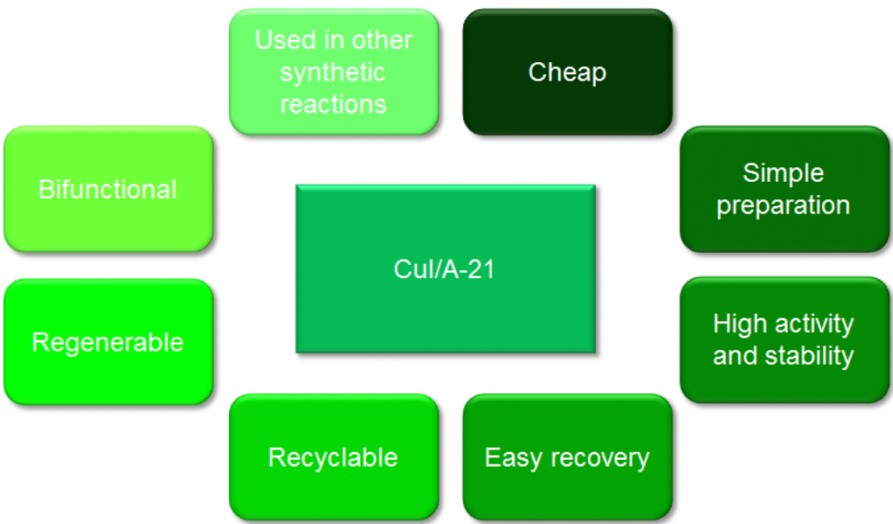

**Figure 13** **Advantages of Amberlyst A-21 supported CuI as a heterogeneous catalyst.**

prevents oxidation or disproportionation of the copper (I) ion, which would affect negatively its activity (*Bosica & Gabarretta, 2015*; *Keshavarz et al., 2013*). Handling of the catalyst is also relatively simple due to the beaded form of the resin and hence, only filtration is required to collect the catalyst after completion of the reaction. Thus, the catalyst was recycled to determine the number of cycles it could be consecutively used. The recycling test was performed on the model reaction for the synthesis of **6**. The catalyst was washed with diethyl ether solvent, once recovered, and dried under vacuum for one night. The catalyst was immediately reused on the following day for the next recycling trial.

As can be seen (Fig. 14), the catalyst had a consistent performance over eight cycles with only a drop of 10% in yield. This suggests that the heterogeneous catalyst has a good activity and stability. Quantitative analysis of the stability of the heterogeneous catalyst was performed by atomic absorption spectroscopy (AAS), to determine the amount of copper that leached in the solution throughout each cycle (Fig. 15). The recycling test was repeated using the same batch of catalyst, 1 mL solution was collected from each cycle and tested to determine the percentage of copper from the catalyst that leached into the solution.

The results (Fig. 15) show that Amberlyst A-21 supported CuI is very stable since only 1% Cu had leached into the solution after eight cycles. Another advantage of the heterogeneous catalyst is that regeneration with more copper (I) iodide is easy, especially in cases where leaching is high, by mixing the polymeric resin support with more metal salt. This increases the shelf-life of the catalyst. The supported catalyst is also advantageous in that it acts as a bifunctional catalyst since only a few of the terminal nitrogen sites in the dimethylamino moiety of the Amberlyst A-21 support are chelated with the Lewis acid copper(I) iodide during mixing (*Girard et al., 2006*). This means that free base moieties are also present and thus both acidic and basic functionalities are active in this catalyst. This also implicates that the catalyst can catalyse different synthetic protocols.

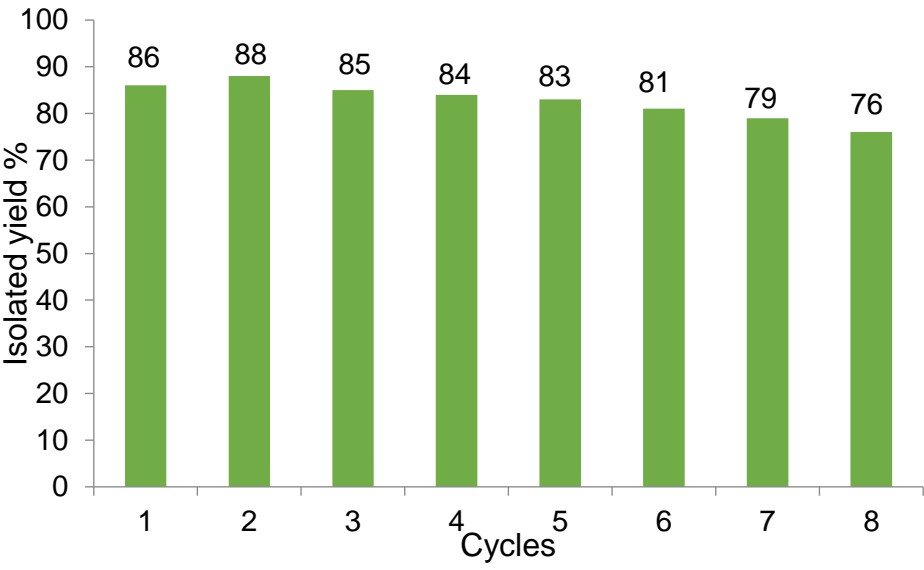

**Figure 14** Recycling test of CuI.A-21 for the synthesis of 6 in the model reaction.

| Cycle | % Cu in solution |
|-------|------------------|
| 1 | 0.007 |
| 2 | 0.36 |
| 3 | 0.21 |
| 4 | 0.095 |
| 5 | 0.0002 |
| 6 | 0.049 |
| 7 | 0.15 |
| 8 | 0.13 |
| | Total=1.00% |

**Figure 15** Percentage of copper ions that leached into the solution from the heterogeneous catalyst CuI.A-21 throughout each cycle.

## CONCLUSIONS

In summary, an efficient one-pot three-component nitro-Mannich reaction between aldehydes, amines and nitroalkanes using CuI on Amberlyst A-21 as heterogeneous catalyst was developed. The reaction exhibited a green protocol with a high atom economy of 93% and a low E-Factor of 1.26 and afforded a wide scope of β-nitroamines, with more than 25 novel products, in moderate to excellent yields. A degree of selectivity was also observed towards one of the diastereomers. The catalyst proved to be highly active due to its bifunctional Lewis acidic and Brønsted basic nature, and recycling tests indicated a consistent performance over eight cycles, in which a drop of 10% in yield was observed. The stability of the catalyst was demonstrated using atomic absorption spectroscopy, showing that only 1% Cu had leached in the solution after eight cycles. The wide versatility in the substrate functional groups used offers the possibility to synthesise compounds that are more complex and to develop novel synthetic routes to natural products, pharmaceutical products, or dyes in the chemical, food and textile industries.

### Experimental
#### General

All reagents are commercially available (Aldrich, Acros, Alfa Aesar, BDH, Cambrian, Fischer Scientific, Hopkins & Williams, Labkem, Macherey-Nagel, Merck, Riedel de Haen, Scharlau) and were used without further purification except for molecular sieves 4 Å, which were heated to 110 °C, and Amberlyst A-21, which was dried before use. TLC plates made up of silica gel on PET-foil with a fluorescent indicator, fluorescing at 254 nm under UV irradiation, were used as a stationary phase for TLC. TLC plates were developed using different mixtures of *n*-hexane and ethyl acetate as the mobile phase. Once developed, the plates were observed under a UV lamp at a wavelength of 254 nm, followed by treatment into an iodine-saturated chamber or oxidised in potassium permanganate. A Shimadzu GC-2010® gas chromatography instrument was used to monitor the imine formation, where possible, and to identify the components of the crude mixture. The equipment consisted of flame ionisation detector and a HiCap5 column with 30 m (length) by 0.32 mm (internal diameter) by 0.25 μm (film thickness) dimensions as the stationary phase. The mobile phase consisted of nitrogen carrier gas. The injection temperature was set at 65–300 °C at a heating rate of 15 °C min$^{-1}$. $^1$H and $^{13}$C NMR spectra were obtained using a Bruker Avance III HD® NMR Spectrometer, containing an Ascend 500 11.75 Tesla Superconducting Magnet, operating at 500.13 MHz and 125.76 MHz for $^1$H and $^{13}$C, respectively. The spectra of the products were recorded using a Multinuclear 5 mm PABBO Probe. ACD/Chemsketch (Freeware) Version 12.01 software was used to interpret the spectra. IR spectroscopy was carried out using a Shimadzu IR Affinity-1 FTIR Spectrometer. Calibration of this instrument was done against a 1,602 cm-1 polystyrene absorbance spectrum. Product mass spectra were generated using a Waters® ACQUITY® TQD system with a tandem quadrupole mass spectrometer. Melting points of solid products were determined using a Gallenkamp melting point apparatus fitted with a mercury thermometer. Flame atomic absorption spectroscopy (FAAS) was performed using a contraAA® 700 High Resolution Continuum Source Atomic Absorption Spectrometer.

### General method for the multicomponent nitro-Mannich reaction

The general procedure for the nitro-Mannich reaction was based using the optimal conditions selected in Fig. 5, entry **l**. To a 25 mL 2-necked round-bottomed flask, equipped with reflux condenser, 5 mol% CuI on Amberlyst A-21 (with a loading of 1.52 mmol CuI $g^{-1}$), aldehyde (2.5 mmol) and amine (2.5 mmol) were introduced via a plastic funnel, and stirred at 100 °C using an oil bath till imine formation was complete. The reaction was followed with using TLC and GC, where possible, at several time intervals. Nitromethane (12.5 mmol) was then added to the same flask and the reaction was monitored by TLC. Upon completion, the reaction was stopped and allowed to cool down to room temperature, where heating was performed, and the catalyst was separated by filtration, using 10 mL ethyl acetate. The crude reaction was then purified by column chromatography (hexane/ethyl acetate) or by direct recrystallization of the reaction mixture from ethanol, and dried under vacuum. Isolated yields were obtained and the products were analysed by IR, $^{1}$H and $^{13}$C NMR spectroscopy, and mass spectrometry.

## Preparation of Copper(I) Iodide on Amberlyst A-21

A sample of 20 g of dry Amberlyst A-21 was first prepared by suspending it in 100 mL methanol for 30 min. After this, the mixture was filtered and washed in methanol for a further three times. This method was repeated in dichloromethane. The resin was then dried on a rotary evaporator at 50 °C until it was free-flowing and subsequently left overnight in a desiccator under vacuum. The dry weight of Amberlyst A-21 was 9.63 g. Amberlyst-21 (5 g) were then added to a solution of 2.86 g (15 mmol) copper(I) iodide in 75 mL acetonitrile and left to stir for 24 h at room temperature. The solvent was then evaporated and the green resin was washed with two 75 mL aliquots of acetonitrile, followed by two 75 mL aliquots of dichloromethane. The resin was then dried on a rotary evaporator at 40 °C and subsequently left overnight in a desiccator under vacuum. The loading of copper per gram of resin was calculated by observing the weight increases of the dried sample of CuI on Amberlyst A-21. A weight increase of 2.05 g (10.8 mmol CuI) was observed, which gave a polymer loading of 1.52 mmol CuI $g^{-1}$.

### Analytical Data for selected products

**6** *N*-(2-nitro-1-phenylethyl)aniline (*Choudhary & Peddinti, 2011*). Brown solid. m.p. 73-75 °C. IR (neat KBr, cm$^{-1}$): $\nu = 3354, 3078, 3058, 3027, 2970, 2916, 2894, 1604, 1546, 1514, 1508, 1489, 1458, 1425, 1381, 1314, 1264, 1213, 1181, 1127, 1083, 1067, 1032, 994, 940, 879, 851, 822, 756, 727, 701, 654, 628, 533, 511$. $^{1}$H NMR (CDCl$_3$, 500 MHz) $\delta$ (ppm): 4.35 (d, 1H, Ha, $J = 6.4$ Hz), 4.70 (d, 2H, Hb, $J = 6.8$ Hz), 5.15 (q, 1H, Hc, $J = 6.8$ Hz), 6.60 (d, 2H, Hd, $J_o = 8.6$ Hz, $J_m = 1$ Hz), 6.75 (t, 1H, He, $J = 7.4$ Hz), 7.15 (t, 2H, Hf, $J = 8.0$ Hz), 7.30 (m,1H Hg), 7.40 (m, 4H Hh, Hi). $^{13}$C NMR (CDCl$_3$, 126 MHz) $\delta$ (ppm): 56.7, 80.0, 113.9, 119.0, 126.5, 128.7, 129.3, 129.4, 137.8, 145.7. MS (ES+) m/z = 242.07 [MH$^+$], 122.03, 94.06, 92.04, 86.98, 92.04.

**7a** *N*-(2-nitro-1-(*p*-tolyl)ethyl)aniline (*Yang, Xia & Huang, 2011*). Dark-red thick oil. IR (neat NaCl, cm$^{-1}$): $\nu = 3400, 3091, 3053, 3027, 2954, 2921, 2863, 1633, 1603, 1554, 1506, 1458, 1436, 1423, 1378,1338, 1316, 1266, 1214, 1182, 1157, 1113, 1068, 1031, 1021, 994,$

971, 914, 872, 814, 752, 693, 669, 510. $^1$H NMR (CDCl$_3$, 500 MHz) $\delta$ (ppm): 2.33 (s, 3H, Ha), 4.35 (d, 1H, Hb, $J = 6.1$ Hz), 4.70 (m, 2H, Hc), 5.15 (q, 1H, Hd, $J = 6.7$ Hz), 6.61 (d, 2H, He, $J = 8.6$ Hz), 6.75 (t, 1H, Hf, $J = 7.4$ Hz), 7.14 (t, 2H, Hg, $J_o = 8.0$ Hz, $J_m = 1.2$ Hz), 7.18 (d, 2H, Hh, $J = 7.9$ Hz), 7.25 (d, 2H, Hi, $J = 8.2$ Hz). $^{13}$C NMR (CDCl$_3$, 126 MHz) $\delta$ (ppm): 21.1, 56.5, 80.0, 113.9, 118.9, 126.4, 129.4, 130.0, 134.7, 138.5, 145.8. MS (ES+) m/z = 257.10 [MH$^+$], 196.12, 164.05, 118.06, 94.06, 91.07.

**7c** *N*-(1-(4-bromophenyl)-2-nitroethyl)aniline. Dark-red thick oil. IR (neat NaCl, cm$^{-1}$): $\nu = 3404$, 3087, 3053, 3026, 2963, 2918, 1603, 1553, 1508, 1423, 1404, 1377, 1335, 1314, 1263, 1215, 1180, 1155, 1128, 1099, 1072, 1011, 993, 902, 918, 876, 822, 752, 692, 648, 540, 509. $^1$H NMR (CDCl$_3$, 500 MHz) $\delta$ (ppm): 4.40 (d, 1H, Ha, $J = 6.3$ Hz), 4.69 (d, 2H, Hb, $J = 6.6$ Hz), 5.12 (q, 1H, Hc, $J = 6.6$ Hz), 6.58 (d, 2H, Hd, $J = 7.7$ Hz), 6.75 (t, 1H, He, $J = 7.4$ Hz), 7.15 (t, 2H, Hf $J = 7.4$ Hz), 7.28 (d, 2H, Hg, $J = 8.5$ Hz), 7.51 (d, 2H, Hh, $J = 8.5$ Hz ). $^{13}$C NMR (CDCl$_3$, 126 MHz) $\delta$ (ppm): 56.1, 79.7, 113.9, 119.2, 122.6, 128.2, 129.4, 132.5, 137.8, 145.3. MS (ES+) m/z = 323.04 [MH$^+$], 262.04, 183.99, 103.02, 93.98, 77.01.

**7d** *N*-(2-nitro-1-(2-nitrophenyl)ethyl)aniline (*Tajbakhsh, Farhang & Hosseini, 2014*). Yellow-brown solid. m.p. 77-79 °C. IR (neat, KBr, cm$^{-1}$): $\nu = 3375$, 3104, 3078, 3030, 2920, 2862, 1622, 1580, 1560, 1522, 1474, 1435, 1386, 1340, 1289, 1263, 1250, 1215, 1182, 1154, 1131, 1086, 1066, 937, 921, 879, 860, 851, 811, 792, 750, 724, 708, 673, 647, 627, 560, 517, 505. $^1$H NMR (CDCl$_3$, 500 MHz) $\delta$ (ppm): 4.83 (m, 2H, Ha), 5.01 (dd, 1H, Hb $J = 3.7$ Hz, 12.4 Hz), 5.80 (m, 1H, Hc), 6.48 (d, 2H, Hd, $J_o = 8.6$ Hz, $J_m = 1.0$ Hz), 6.75 (t, 1H, He, $J_o = 7.4$ Hz, $J_m = 1.0$ Hz), 7.10 (t, 2H, Hf, $J = 8.0$ Hz), 7.50 (t, 1H, Hg, $J_o = 8.1$ Hz, $J_m = 1.4$ Hz), 7.60 (t, 1H, Hh, $J_o = 7.5$ Hz, $J_m = 1.2$ Hz), 7.69 (d, 1H, Hi, $J_o = 7.9$ Hz, $J_m = 1.3$ Hz), 8.09 (d, 1H, Hj, $J_o = 8.2$ Hz, $J_m = 1.2$ Hz).

**7e** 3-(2-Nitro-1-(phenylamino)ethyl)phenol. Dark-red oil. IR (neat NaCl, cm$^{-1}$): $\nu = 3407$, 3097, 3072, 3036, 2923, 2856, 1613, 1580, 1560, 1509, 1451, 1422, 1380, 1344, 1325, 1289, 1273, 1257, 1215, 1186, 1154, 1138, 1083, 1066, 1028, 996, 970, 934, 918, 876, 860, 844, 792, 750, 718, 692, 669, 643, 611, 518. $^1$H NMR (CDCl$_3$, 500 MHz) $\delta$ (ppm): 4.38 (m, 1H, Ha), 4.69 (d, 2H, Hb, $J = 6.7$ Hz), 5.01 (broad, 1H, Hc), 5.11 (q, 1H, Hd, $J = 6.1$ Hz), 6.60 (d, 2H, He, $J = 8.6$ Hz), 6.75 (t, 1H, Hf, $J = 7.4$ Hz), 6.76 (m, 1H, Hg), 6.85 (s, 1H, Hh), 6.95 (d, 1H, Hi, $J = 7.6$ Hz), 7.15 (t, 2H, Hj, $J_o = 7.0$ Hz, $J_m = 1.2$ Hz), 7.25 (t, 1H, Hk, $J = 7.9$ Hz). $^{13}$C NMR (CDCl$_3$, 126 MHz) $\delta$ (ppm): 56.4, 79.9, 113.4, 113.9, 115.7, 118.6, 119.0, 129.4, 130.6, 139.6, 145.6, 156.4. MS (ES+) m/z = 259.10 [MH$^+$], 212.17, 198.06, 166.06, 120.07, 94.05.

**7g** *N*-(1-([1,1-biphenyl]-4-yl)-2-nitroethyl)aniline. Yellow-red solid. m.p. 97-99 °C. IR (neat, KBr, cm$^{-1}$): $\nu = 3398$, 3055, 3030, 2960, 2918, 1603, 1549, 1511, 1486, 1476, 1460, 1441, 1422, 1381, 1314, 1266, 1257, 1213, 1180, 1155, 1133, 1076, 1008, 993, 920, 876, 837, 768, 755, 732, 695, 650, 621, 561, 507. $^1$H NMR (CDCl$_3$, 500 MHz) $\delta$ (ppm): 4.42 (d, 1H, Ha, $J = 6.4$ Hz), 4.75 (d, 2H, Hb, $J = 6.8$ Hz), 5.22 (q, 1H, Hc, $J = 6.6$ Hz), 6.65 (d, 2H, Hd, $J_o = 8.6$ Hz, $J_m = 1.0$ Hz), 6.75 (t, 1H, He, $J_o = 7.4$ Hz, $J_m = 1.0$ Hz), 7.16 (t, 2H, Hf, $J_o = 8.0$ Hz, $J_m = 1.2$ Hz), 7.35 (t, 1H, Hg, $J_o = 7.4$ Hz, $J_m = 1.3$ Hz), 7.43 (t, 2H, Hh, $J = 7.6$ Hz), 7.47 (d, 2H, Hi, $J_o = 8.1$ Hz, $J_m = 1.6$ Hz), 7.55 (d, 2H, Hj, $J_o = 8.2$ Hz, $J_m = 2.0$ Hz), 7.60 (d, 2H, Hk, $J_o = 8.4$ Hz, $J_m = 2.0$ Hz). $^{13}$C NMR (CDCl$_3$, 126 MHz) $\delta$ (ppm):

56.5, 79.9, 114.0, 119.0, 126.9, 127.1, 127.6, 128.0, 128.9, 129.4, 136.7, 140.3, 141.7, 145.7. MS (ES+) m/z = 327.03 [MH$^+$], 80.91.

**7h** *N*-(1-(3,4-dichlorophenyl)-2-nitroethyl)aniline. Dark-red thick oil. IR (neat NaCl, cm$^{-1}$): $\nu$ = 3403, 3093, 3054, 3025, 2960, 2922, 2857, 1604, 1556, 1508, 1472, 1423, 1378, 1313, 1262, 1214, 1194, 1181, 1136, 1075, 1033, 997, 929, 884, 826, 752, 697, 668, 619, 584. $^1$H NMR (CDCl$_3$, 500 MHz) $\delta$ (ppm): 4.43 (broad, 1H, Ha), 4.68 (d, 2H, Hb, *J* = 6.9 Hz), 5.11 (q, 1H, Hc, *J* = 6.0 Hz), 6.57 (d, 2H, Hd, $J_o$ = 9.1 Hz, $J_m$ = 0.95 Hz), 6.78 (t, 1H, He, $J_o$ = 7.4 Hz, $J_m$ = 1.0 Hz), 7.15 (t, 2H, Hf, *J* = 8.0 Hz), 7.25 (d, 1H, Hg, *J* = 9.3 Hz), 7.45 (d, 1H, Hh, *J* = 8.2 Hz), 7.51 (s, 1H, Hi). $^{13}$C NMR (CDCl$_3$, 126 MHz) $\delta$ (ppm): 55.8, 79.6, 114.0, 119.5, 125.8, 128.6, 129.5, 131.6, 132.9, 133.6, 138.1, 145.1. MS (ES+) m/z = 312.97 [MH$^+$], 252.96, 252.02, 173.91, 101.73, 93.98.

**7i** *N*-(1-(2,6-dichlorophenyl)-2-nitroethyl)aniline. Red solid. m.p. 96–99 °C. IR (neat, KBr, cm$^{-1}$): $\nu$ = 3388, 3061, 3049, 3033, 2982, 2928, 2855, 1603, 1581, 1555, 1508, 1495, 1457, 1441, 1381, 1339, 1311, 1289, 1266, 1250, 1216, 1200, 1181, 1152, 1123, 1089, 1056, 1028, 993, 923, 901, 885, 860, 834, 793, 774, 758, 726, 695, 640, 621, 558, 495. $^1$H NMR (CDCl$_3$, 500 MHz) $\delta$ (ppm): 4.73 (dd, 1H, Ha, *J* = 5.2 Hz, 12.3 Hz), 4.95 (d, 1H, Hb, 11.2 Hz), 5.07 (dd, 1H, Hc, 9.9 Hz, 12.3 Hz), 6.29 (ddd, 1H, Hd, 11.2 Hz, 10.0 Hz, 5.3 Hz), 6.74 (m, 3H, He, Hf), 7.13-7.19 (m, 3H, Hg, Hh), 7.27-7.38 (m, 2H, Hi). $^{13}$C NMR (CDCl$_3$, 126 MHz) $\delta$ (ppm): 53.3, 76.5, 114.2, 119.4, 129.4, 130.2, 132.2, 145.3. MS (ES+) m/z = 310.97 [MH$^+$], 250.02, 136.98, 124.95, 93.33.

**7j** *N*-(1-(benzo[*d*][1,3]dioxol-5-yl)-2-nitroethyl)aniline. Red thick oil. IR (neat NaCl, cm$^{-1}$): $\nu$ = 3393, 3051, 3028, 2960, 2896, 1607, 1559, 1508, 1491, 1449, 1378, 1339, 1313, 1246, 1184, 1155, 1110, 1036, 997, 971, 929, 871, 816, 755, 733, 693, 668, 639. $^1$H NMR (CDCl$_3$, 500 MHz) $\delta$ (ppm): 4.35 (broad, 1H, Ha), 4.66 (d, 2H, Hb, *J* = 6.7 Hz), 5.07 (q, 1H, Hc, *J* = 6.0 Hz), 5.96 (m, 2H, Hd), 6.61 (d, 2H, He, $J_o$ = 8.6 Hz, $J_m$ = 1.0 Hz), 6.75 (t, 1H, Hf, *J* = 7.3 Hz), 6.79 (d, 1H, Hg, *J* = 8.6 Hz), 6.87 (m, 2H, Hh, Hi), 7.15 (t, 2H, Hj, $J_o$ = 7.4 Hz, $J_m$ = 2.0 Hz). $^{13}$C NMR (CDCl$_3$, 126 MHz) $\delta$ (ppm): 56.5, 80.2, 101.4, 106.7, 108.9, 113.9, 119.0, 120.0, 129.4, 131.6, 145.6, 147.9, 148.5. MS (ES+) m/z = 287.10 [MH$^+$], 194.02, 148.03, 94.05, 91.64, 90.99.

**7k** *N*-(1-nitropentan-2-yl)aniline. Red oil. IR (neat NaCl, cm$^{-1}$): $\nu$ = 3367, 3055, 3017, 2965, 2931, 2875, 1684, 1604, 1555, 1500, 1463, 1445, 1383, 1362, 1317, 1258, 1220, 1185, 1140, 1075, 1047, 994, 922, 839, 756, 692, 669. $^1$H NMR (CDCl$_3$, 500 MHz) $\delta$ (ppm): 0.95 (t, 3H, Ha, *J* = 7.3 Hz), 1.40–1.70 (m, 4H, Hb, Hc), 3.65 (d, 1H, Hd, *J* = 7.2 Hz), 4.10 (m, 1H, He), 4.40 (dd, 1H, Hf, *J* = 6.2 Hz, 11.8 Hz), 4.50, (dd, 1H, Hg, *J* = 5.1 Hz, 11.8 Hz), 6.65 (d, 2H, Hh, $J_o$ = 8.6 Hz, $J_m$ = 0.9 Hz), 6.75 (t, 1H, Hi, $J_o$ = 7.4 Hz, $J_m$ = 0.9 Hz), 7.20 (t, 2H, Hj, $J_o$ = 8.0 Hz, $J_m$ = 1.2 Hz). $^{13}$C NMR (CDCl$_3$, 126 MHz) $\delta$ (ppm): 13.8, 19.1, 35.1, 52.0, 78.1, 113.5, 118.7, 129.6, 146.0. MS (ES+) m/z = 209.10 [MH$^+$], 148.11, 118.25, 106.02, 93.08, 40.98.

**7l** *N*-(1-cyclohexyl-2-nitroethyl)aniline. Red oil. (neat NaCl, cm$^{-1}$): $\nu$ = 3399, 3053, 3022, 2930, 2853, 1601, 1557, 1504, 1449, 1427, 1381, 1348, 1315, 1256, 1215, 1180, 1155, 1123, 1072, 1030, 993, 964, 914, 893, 874, 841, 752, 692, 667, 619, 509. $^1$H NMR (CDCl3, 500 MHz) $\delta$: 1.00–1.30 (m, 5H, Ha, Hb, Hc), 1.55-1.85 (m, 5H, Hd, He, Hf), 1.91 (d, 1H, Hg, *J* = 12.5 Hz), 3.70 (d, 1H, Hh, *J* = 9.8 Hz), 3.94–4.00 (m, 1H, Hi), 4.45 (dd, 1H, Hj

$J = 6.3$ Hz, 12.1 Hz), 4.55 (dd, 1H, Hk, $J = 5.7$ Hz, 12.1 Hz), 6.65 (d, 2H, Hl, $J = 7.7$ Hz), 6.75 (t, 1H, Hm, $J = 7.3$ Hz), 7.2 (t, 2H, Hn, $J_o = 7.9$ Hz, $J_m = 1.0$ Hz). $^{13}$C NMR (CDCl$_3$, 126 MHz) $\delta$ (ppm): 26.0, 26.1, 28.3, 29.6, 40.6, 57.0, 76.5, 113.5, 118.4, 129.5, 146.6. MS (ES+) m/z $= 249.14$ [MH$^+$], 188.17, 109.08, 106.02, 94.71, 67.06.

**7m** Ethyl 3-nitro-2-(phenylamino)propanoate. Red oil. IR (neat NaCl, cm$^{-1}$): $\nu = 3388$, 3054, 3028, 2983, 2936, 2905, 2870, 1742, 1605, 1558, 1507, 1447, 1421, 1380, 1313, 1281, 1256, 1215, 1158, 1100, 1078, 1053, 1018, 958, 922, 875, 856, 757, 693. $^1$H NMR (CDCl$_3$, 500 MHz) $\delta$ (ppm): 1.29 (t, 3H, Ha, $J = 7.1$ Hz), 4.29 (m, 2H, Hb), 4.51 (d, 1H, Hc, $J = 7.7$ Hz), 4.64 (m, 1H, Hd), 4.79 (dd, 1H, He, $J = 5.0$ Hz, 13.7 Hz), 4.87 (dd, 1H, Hf, $J = 4.5$ Hz, 13.7 Hz), 6.68 (d, 2H, Hg, $J = 7.7$ Hz), 6.84 (t, 1H, Hh, $J = 7.4$ Hz), 7.22 (t, 2H, Hi, $J_o = 8.0$ Hz, $J_m = 1.9$ Hz). $^{13}$C NMR (CDCl$_3$, 126 MHz) $\delta$ (ppm): 14.0, 54.9, 62.7, 75.6, 113.8, 119.7, 129.6, 145.2, 169.5. MS (ES+) m/z $= 239.10$ [MH$^+$], 178.11, 150.07, 104.02, 93.02, 77.02.

**8c** 3-Methyl-*N*-(2-nitro-1-phenylethyl)aniline. Red thick oil. IR (neat NaCl, cm$^{-1}$): $\nu = $ 3396, 3027, 2949, 2919, 2859, 1608, 1591, 1558, 1508, 1491, 1453, 1424, 1380, 1339, 1323, 1269, 1215, 1178, 1126, 1078, 1032, 998, 970, 926, 847, 768, 699, 668, 624, 592, 527. $^1$H NMR (CDCl$_3$, 500 MHz) $\delta$ (ppm): 2.24 (s, 3H, Ha), 4.31 (d, 1H, Hb, $J = 6.4$ Hz), 4.71 (d, 2H, Hc, $J = 6.8$ Hz), 5.17 (q, 1H, Hd, $J = 6.7$ Hz), 6.41 (d, 1H, He, $J_o = 8.0$ Hz, $J_m = 2.2$ Hz), 6.46 (s, 1H, Hf), 6.58 (d, 1H, Hg, $J = 7.5$ Hz), 7.03 (t, 1H, Hh, $J = 7.8$ Hz), 7.31–7.34 (m, 1H, Hi), 7.35–7.41 (m, 4H, Hj, Hk). $^{13}$C NMR (CDCl$_3$, 126 MHz) $\delta$ (ppm): 21.5, 56.6, 79.9, 110.9, 114.8, 119.9, 126.4, 128.6, 129.2, 129.3, 137.8, 139.2, 145.7. MS (ES+) m/z $= 257.08$ [MH $^+$], 196.11, 149.99, 107.98, 104.07, 78.05.

**8d** 2-Methyl-*N*-(2-nitro-1-phenylethyl)aniline. Red thick oil. IR (neat NaCl, cm$^{-1}$): $\nu = $ 3417, 3058, 3024, 2967, 2915, 2853, 1636, 1608, 1588, 1554, 1508, 1481, 1455, 1423, 1377, 1345, 1313, 1263, 12161189, 1160, 1136, 1074, 1053, 1030, 1003, 987, 968, 923, 844, 751, 701, 653, 621, 592, 536. $^1$H NMR (CDCl$_3$, 500 MHz) $\delta$ (ppm): 2.23 (s, 3H, Ha), 4.35 (d, 1H, Hb, $J = 5.5$ Hz), 4.75 (d, 2H, Hc, $J = 6.7$ Hz), 5.19 (q, 1H, Hd, $J = 6.4$ Hz), 6.46 (d, 1H, He, $J = 8.0$ Hz), 6.69 (t, 1H, Hf, $J_o = 7.3$ Hz, $J_m = 0.8$ Hz), 7.00 (t, 1H, Hg, $J_o = 7.5$ Hz, $J_m = 1.0$ Hz), 7.08 (d, 1H, Hh, $J = 7.4$ Hz), 7.31–7.34 (m, 1H, Hi), 7.36–7.42 (m, 4H, Hj, Hk). $^{13}$C NMR (CDCl$_3$, 126 MHz) $\delta$ (ppm): 17.5, 56.8, 80.3, 111.2, 118.5, 123.0, 126.4, 127.1, 128.7, 129.4, 130.4, 137.9, 143.7. MS (ES+) m/z $= 257.08$ [MH$^+$], 196.11, 149.99, 107.98, 104.07, 77.99.

**8e** 4-Ethyl-*N*-(2-nitro-1-phenylethyl)aniline. Red thick oil. IR (neat NaCl, cm$^{-1}$): $\nu = $ 3400, 3062, 3027, 2962, 2928, 2889, 2870, 1615, 1557, 1521, 1494, 1452, 1426, 1414, 1377, 1343, 1313, 1265, 1212, 1184, 1134, 1113, 1076, 1029, 1000, 968, 921, 826, 767, 700, 644, 628, 540. $^1$H NMR (CDCl$_3$, 500 MHz) $\delta$ (ppm): 1.15 (t, 3H, Ha, $J = 7.6$ Hz), 2.50 (q, 2H, Hb, $J = 7.6$ Hz), 4.27 (broad, 1H, Hc), 4.70 (d, 2H, Hd, $J = 6.8$ Hz), 5.14 (broad, 1H, He), 6.55 (d, 2H, Hf, $J = 8.5$ Hz), 6.98 (d, 2H, Hg, $J = 8.5$ Hz), 7.32 (m, 1H, Hh), 7.36–7.41 (m, 4H, Hi, Hj). $^{13}$C NMR (CDCl$_3$, 126 MHz) $\delta$ (ppm): 15.8, 27.9, 57.0, 80.0, 114.1, 126.5, 128.6, 128.7, 129.3, 134.8, 138.0, 143.6. MS (ES+) m/z $= 271.08$ [MH$^+$], 210.11, 121.37, 104.01, 103.69, 78.06.

**8h** 1-(4-((2-Nitro-1-phenylethyl)amino)phenyl)ethanone. Dark-red thick oil. IR (neat NaCl, cm$^{-1}$): $\nu = 3319$, 3156, 3087, 3068, 3027, 2963, 2919, 2864, 1651, 1594, 1562, 1533,

1496, 1457, 1425, 1378, 1362, 1356, 1283, 1216, 1181, 1136, 1117, 1073, 1025, 955, 926, 834, 812, 765, 724, 704, 638, 599, 577. $^1$H NMR (CDCl$_3$, 500 MHz) $\delta$ (ppm): 2.47 (s, 3H, Ha), 4.75 (m, 2H, Hb), 4.99 (d, 1H, Hc, $J = 6.7$ Hz), 5.26 (q, 1H, Hd, $J = 7.3$ Hz), 6.60 (d, 2H, He, $J_o = 6.9$ Hz, $J_m = 1.9$ Hz), 7.33-7.42 (m, 5H, Hf, Hg, Hh), 7.79 (d, 2H, Hi, $J_o = 6.9$ Hz, $J_m = 1.9$ Hz). $^{13}$C NMR (CDCl$_3$, 126 MHz) $\delta$ (ppm): 26.1, 56.0, 79.8, 112.7, 126.4, 128.2, 129.0, 129.5, 130.7, 136.8, 149.8, 196.5. MS (ES+) m/z = 285.10 [MH$^+$], 224.14, 135.09, 104.07, 93.08, 42.99.

**8i** 4-((2-Nitro-1-phenylethyl)amino)benzonitrile. Red thick oil. IR (neat NaCl, cm$^{-1}$): $\nu = 3337, 3059, 3027, 3005, 2965, 2915, 2843, 2215, 1605, 1558, 1519, 1494, 1458, 1417, 1377, 1343, 1311, 1278, 1217, 1179, 1139, 1102, 1074, 1028, 1001, 968, 920, 825, 761, 700, 668, 627, 606, 546. $^1$H NMR (CDCl$_3$, 500 MHz) $\delta$ (ppm): 4.71 (dd, 1H, Ha, $J = 8.5$ Hz, 12.5 Hz), 4.76 (dd, 1H, Hb, $J = 4.7$ Hz, 12.5 Hz), 5.03 (broad d, 1H, Hc, $J = 6.5$ Hz), 5.20 (m, 1H Hd), 6.60 (d, 2H, He, $J_o = 8.9$ Hz, $J_m = 2.0$ Hz), 7.34–7.38 (m, 3H, Hf, Hg), 7.38–7.42 (m, 4H, Hh, Hi). $^{13}$C NMR (CDCl$_3$, 126 MHz) $\delta$ (ppm): 55.9, 79.7, 100.7, 113.5, 120.0, 126.3, 129.1, 129.6, 133.8, 136.9, 149.2. MS (ES+) m/z = 268.10 [MH$^+$], 221.12, 143.08, 118.10, 91.05.

**8j** 3-Nitro-$N$-(2-nitro-1-phenylethyl)aniline. Red thick oil. IR (neat NaCl, cm$^{-1}$): $\nu = 3392, 3085, 3066, 3030, 3004, 2971, 2921, 2862, 1623, 1561, 1526, 1507, 1459, 1424, 1379, 1348, 1276, 1220, 1129, 1099, 1081, 998, 964, 890, 853, 817, 793, 761, 737, 700, 670, 624, 529. $^1$H NMR (CDCl$_3$, 500 MHz) $\delta$ (ppm): 4.70-4.75 (dd, 1H, Ha, $J = 8.7$ Hz, 12.4 Hz), 4.75–4.80 (dd, 1H, Hb, $J = 4.5$ Hz, 12.4 Hz), 4.88 (d, 1H, Hc, $J = 6.7$ Hz), 5.23 (d, 1H, Hd, $J = 4.7, 6.6, 8.5$ Hz), 6.89 (d, 1H, He, $J_o = 8.2$ Hz, $J_m = 2.0$ Hz), 7.25 (t, 1H, Hf, $J = 8.2$ Hz), 7.34–7.36 (m, 1H, Hg), 7.38–7.42 (m, 5H, Hh, Hi, Hj), 7.55 (d, 1H, Hk $J_o = 8.2$ Hz, $J_m = 1.4$ Hz). $^{13}$C NMR (CDCl$_3$, 126 MHz) $\delta$ (ppm): 56.4, 79.9, 108.0, 113.5, 119.6, 126.3, 129.1, 129.6, 130.0, 136.4, 146.6, 149.3. MS (ES+) m/z = 288.08 [MH$^+$], 150.05, 138.99, 104.06, 91.97, 77.99.

**8k** 2,5-Dichloro-$N$-(2-nitro-1-phenylethyl)aniline. Red thick oil. IR (neat NaCl, cm$^{-1}$): $\nu = 3483, 3396, 3090, 3067, 3031, 2963, 2921, 2853, 1594, 1559, 1508, 1485, 1459, 1417, 1378, 1343, 1294, 1211, 1136, 1094, 1046, 952, 910, 835, 794, 768, 700, 613, 581, 532. $^1$H NMR (CDCl$_3$, 500 MHz) $\delta$ (ppm): 4.75 (m, 2H, Ha), 5.18 (q, 1H, Hb, $J = 6.4$ Hz), 5.23 (d, 1H, Hc, $J = 6.9$ Hz), 6.51 (s, 1H, Hd, $J_m = 2.3$ Hz), 6.65 (d, 1H, He, $J_o = 8.4$ Hz, $J_m = 2.3$ Hz), 7.18 (d, 1H, Hf, $J = 8.4$ Hz), 7.35-7.42 (m, 5H, Hg, Hh, Hi). $^{13}$C NMR (CDCl$_3$, 126 MHz) $\delta$ (ppm): 56.3, 79.9, 112.4, 118.2, 118.7, 126.3, 129.1, 129.6, 130.0, 133.6, 136.4, 142.5. MS (ES+) m/z = 312.97 [MH$^+$], 252.02, 217.96, 173.91, 101.73, 93.98.

**8l** ($E$)-$N$-(2-nitro-1-phenylethyl)-4-(phenyldiazenyl)aniline. Red solid. m.p. 103–105 °C. IR (neat, KBr, cm$^{-1}$): $\nu = 3483, 3396, 3090, 3067, 3031, 2963, 2921, 2853, 1594, 1559, 1508, 1485, 1459, 1417, 1378, 1343, 1294, 1211, 1136, 1094, 1046, 952, 910, 835, 794, 768, 700, 613, 581, 532. $^1$H NMR (CDCl$_3$, 500 MHz) $\delta$ (ppm): 4.77 (m, 1H, Ha), 4.88 (broad, 1H, Hb), 5.28 (broad, 1H, Hc), 6.69 (d, 2H, Hd, $J_o = 6.8$ Hz, $J_m = 2.1$ Hz), 7.32–7.42 (m, 6H, He, Hf, Hg, Hh), 7.46 (t, 2H, Hi, $J_o = 7.5$ Hz, $J_m = 1.5$ Hz), 7.79 (d, 2H, Hj, $J_o = 6.9$ Hz, $J_m = 2.1$ Hz), 7.82 (d, 2H, Hk, $J = 8.4$ Hz). $^{13}$C NMR (CDCl$_3$, 126 MHz) $\delta$ (ppm): 56.3, 79.8, 113.6, 122.4, 125.1, 126.4, 128.9, 129.0, 129.5, 130.0, 137.0, 145.8, 148.2, 152.9. MS (ES+) m/z = 347.16 [MH$^+$], 197.07, 104.13, 91.97, 77.07, 65.04.

**4a** 3-Nitro-*N*-(2-nitro-1-(*p*-tolyl)ethyl)aniline. Red thick oil. IR (neat NaCl, cm$^{-1}$): $\nu =$ 3398, 3089, 3027, 2923, 2864, 1624, 1588, 1559, 1536, 1422, 1380, 1351, 1266, 1214, 1181, 1109, 1096, 1071, 1044, 1025, 996, 966, 918, 859, 820, 794, 758, 739, 673, 521. $^{1}$H NMR (CDCl$_3$, 500 MHz) $\delta$ (ppm): 2.34 (s, 3a, Ha), 4.72 (m, 2H, Hb), 4.86 (d, 1H, Hc, $J =$ 6.7 Hz), 5.19 (ddd, 1H, Hd, $J =$ 8.4 Hz, 7.0 Hz, 4.7 Hz), 6.89 (d, 1H, He, $J_o =$ 8.2 Hz, $J_m =$ 0.7 Hz), 7.2 (d, 2H, Hf, $J =$ 8.9 Hz), 7.24-7.29 (m, 3H, Hg, Hh), 7.41 (s, 1H, Hi, $J_m =$ 2.3 Hz), 7.56 (d, 1H, Hj, $J_o =$ 8.1 Hz, $J_m =$ 0.8 Hz). $^{13}$C NMR (CDCl$_3$, 126 MHz) $\delta$ (ppm): 21.1, 56.2, 80.0, 108.0, 113.4, 119.6, 126.2, 130.0, 130.2, 133.4, 139.0, 146.7, 149.3. MS (ES+) m/z = 302.10 [MH$^+$], 164.08, 118.10, 121.03, 117.71, 91.05.

**4b** 4-Chloro-*N*-(1-(4-methoxyphenyl)-2-nitroethyl)aniline. Red thick oil. IR (neat NaCl, cm$^{-1}$): $\nu =$ 3390, 3106, 3070, 3034, 3008, 2962, 2936, 2916, 2838, 1605, 1559, 1513, 1497, 1464, 1421, 1379, 1340, 1310, 1254, 1176, 1121, 1091, 1032, 970, 826, 783, 731, 682, 640, 564, 548, 525, 509. $^{1}$H NMR (CDCl$_3$, 500 MHz) $\delta$ (ppm): 3.79 (s, 3H, Ha), 4.4 (d, 1H, Hb, $J =$ 6.4 Hz), 4.67 (d, 2H, Hc, $J =$ 6.7 Hz), 5.06 (q, 1H, Hd, $J =$ 6.7 Hz), 6.52 (d, 2H, He, $J =$ 8.9 Hz), 6.90 (d, 2H, Hf, $J =$ 8.8 Hz), 7.08 (d, 2H, Hg, $J =$ 8.9 Hz), 7.28 (d, 2H, Hh, $J =$ 8.6 Hz). $^{13}$C NMR (CDCl$_3$, 126 MHz) $\delta$ (ppm): 55.3, 56.3, 80.1, 114.8, 115.1, 127.6, 129.2, 131.2, 139.1, 144.3, 159.9. MS (ES+) m/z = 307.09 [MH$^+$], 180.05, 119.04, 91.06.

**4c** 3-(1-((4-methoxyphenyl)amino)-2-nitroethyl)phenol. Dark-red thick oil. IR (neat NaCl, cm$^{-1}$): $\nu =$ 3361, 3306, 3066, 3035, 3000, 2952, 2931, 2827, 1591, 1560, 1513, 1457, 1377, 1298, 1239, 1183, 1127, 1035, 997, 934, 827, 788, 730, 699, 517. $^{1}$H NMR (CDCl$_3$, 500 MHz) $\delta$ (ppm): 3.71 (s, 3H, Ha), 4.66 (d, 2H, Hb, $J =$ 6.7 Hz), 4.74 (m, 1H, Hc), 5.02 (t, 1H, Hd, $J =$ 6.7 Hz), 5.16-5.38 (broad, 1H, He), 6.57 (d, 2H, Hf, $J_o =$ 8.9 Hz, $J_m =$ 2.3 Hz), 6.73 (d, 2H, Hg, $J_o =$ 8.9 Hz, $J_m =$ 2.2 Hz), 6.77 (d, 1H, Hh, $J_o =$ 9.9 Hz, $J_m =$ 0.6 Hz), 6.85 (s, 1H, Hi, $J_m =$ 2.0 Hz), 6.94 (d, 1H, Hj, $J =$ 8.6 Hz, $J_m =$ 1.5 Hz), 7.24 (t, 1H, Hk, $J =$ 7.9 Hz). $^{13}$C NMR (CDCl$_3$, 126 MHz) $\delta$ (ppm): 55.7, 57.5, 80.0, 114.6, 115.0, 115.7, 117.2, 122.4, 130.2, 130.6, 139.7, 139.8, 144.0, 153.0, 156.5. MS (ES+) m/z = 289.07 [MH$^+$], 200.15, 166.06, 123.92.

**4d** 4-bromo-*N*-(1-cyclohexyl-2-nitroethyl)aniline. Red solid. IR m.p. 74–75 °C. IR (neat, KBr, cm$^{-1}$): $\nu =$ 3405, 3386, 3100, 3077, 3064, 3038, 2929, 2851, 1598, 1558, 1509, 1489, 1446, 1424, 1387, 1354, 1318, 1299, 1250, 1217, 1184, 1155, 1135, 1115, 1096, 1079, 1030, 1014, 1000, 971, 931, 915, 892, 816, 803, 745, 699, 672, 639, 597, 508, 442, 423. $^{1}$H NMR (CDCl$_3$, 500 MHz) $\delta$ (ppm): 1.00-1.30 (m, 5H, Ha, Hb, Hc), 1.60 (m, 1H, Hd), 1.65-1.85 (m, He, Hf), 1.90 (m, Hg), 3.72 (d, 1H, Hh, $J =$ 9.8 Hz), 3.92 (m, 1H, Hi), 4.46, (dd, 1H, Hj, $J =$ 6.8 Hz, 12.2 Hz), 4.53 (dd, 1H, Hk, $J =$ 5.2 Hz, 12.2 Hz), 6.53 (d, 2H, Hl, $J =$ 7.4 Hz), 7.26 (m, 2H, Hm). $^{13}$C NMR (CDCl$_3$, 126 MHz) $\delta$ (ppm): 25.9, 26.1, 28.9, 29.6, 40.8, 57.2, 76.6, 110.0, 115.0, 132.3, 145.7. MS (ES+) m/z = 325.03 [MH$^+$], 169.93, 154.07, 136.05, 78.89, 60.02.

**4e** 4-methyl-*N*-(2-nitro-1-(2-nitrophenyl)ethyl)aniline. Yellow-red solid. m.p. 85-87 °C. IR (neat, KBr, cm$^{-1}$): $\nu =$ 3375, 3104, 3078, 3030, 2920, 2862, 1622, 1580, 1560, 1522, 1474, 1435, 1386, 1340, 1289, 1263, 1250, 1215, 1182, 1154, 1131, 1086, 1066, 937, 921, 879, 860, 851, 811, 792, 750, 724, 708, 673, 647, 627, 560, 517, 505. $^{1}$H NMR (CDCl$_3$, 500 MHz) $\delta$ (ppm): 2.18 (s, 3H, Ha), 4.64-4.86 (broad, 1H, Hb), 4.83 (dd, 1H, Hc, $J =$ 7.6 Hz, 12.5 Hz), 5.00 (dd, 1H, Hd, $J =$ 3.7 Hz, 12.5 Hz), 5.76 (dd, 1H, He, $J =$ 3.7 Hz, 7.6 Hz), 6.40 (d, 2H,

Hf, $J = 8.5$ Hz), 6.66 (d, 2H, Hg, $J = 8.5$ Hz), 7.49 (t, 1H, Hh, $J_o = 7.8$ Hz, $J_m = 1.5$ Hz), 7.60 (t, 1H, Hi, $J_o = 7.6$ Hz, $J_m = 1.4$ Hz), 7.68 (d, 1H, Hj, $J_o = 7.8$ Hz, $J_m = 1.4$ Hz), 8.08 (d, 1H, Hk, $J_o = 8.1$ Hz, $J_m = 1.4$ Hz). $^{13}$C NMR (CDCl$_3$, 126 MHz) $\delta$ (ppm): 20.4, 52.9, 79.3, 113.8, 125.7, 128.7, 129.0, 129.6, 130.0, 133.5, 134.4, 142.5, 148.7. MS (ES+) m/z = 302.10 [MH$^+$], 241.15, 120.05, 119.73, 107.05.

**4g** $N$-(1-(4-bromophenyl)-2-nitroethyl)-4-ethylaniline. Red oil. IR (neat NaCl, cm$^{-1}$): $\nu = 3377$, 3086, 3015, 2957, 2919, 2860, 1604, 1582, 1547, 1511, 1478, 1446, 1414, 1369, 1330, 1307, 1259, 1211, 1178, 1126, 1107, 1097, 1069, 1039, 1007, 962, 917, 820, 759, 726, 661, 648, 532. $^1$H NMR (CDCl$_3$, 500 MHz) $\delta$ (ppm): 1.15 (t, 3H, Ha, $J = 7.6$ Hz), 2.50 (q, 2H, Hb, $J = 7.6$ Hz), 4.29 (d, 1H, Hc, $J = 6.1$ Hz), 4.67 (d, 2H, Hd, $J = 6.7$ Hz), 5.09 (q, 1H, He, $J = 6.5$ Hz), 6.52 (d, 2H, Hf, $J_o = 8.5$ Hz, $J_m = 2.0$ Hz), 6.98 (d, 2H, Hg, $J = 8.5$ Hz), 7.28 (d, 2H, Hh, $J_o = 8.4$ Hz, $J_m = 1.7$ Hz), 7.50 (d, 2H, Hi, $J_o = 8.5$ Hz, $J_m = 1.9$ Hz). $^{13}$C NMR (CDCl$_3$, 126 MHz) $\delta$ (ppm): 15.8, 27.9, 56.5, 79.8, 114.1, 122.6, 128.2, 128.8, 132.5, 135.2, 137.0, 143.2. MS (ES+) m/z = 351.08 [MH$^+$], 335.77, 180.26, 90.98, 64.97.

**4h** 4-methyl-N-(1-nitropentan-2-yl)aniline. Red oil. IR (neat NaCl, cm$^{-1}$): $\nu = 3398$, 3027, 2965, 2933, 2871, 1621, 1588, 1553, 1520, 1468, 1429, 1383, 1354, 1321, 1305, 1259, 1230, 1208, 1188, 1155, 1123, 1038, 993, 921, 813, 735, 644, 628, 511. $^1$H NMR (CDCl$_3$, 500 MHz) $\delta$ (ppm): 0.95 (t, 3H, Ha, $J = 7.3$ Hz), 1.38–1.68 (m, 4H, Hb, Hc), 2.24 (s, 3H, Hd), 3.52 (d, 1H, He, $J = 7.9$ Hz), 4.03 (q, 1H, Hf, $J = 5.8$ Hz), 4.41 (dd, 1H, Hg, $J = 6.1$ Hz, 11.7 Hz), 4.52 (dd, 1H, Hh, $J = 5.0$ Hz, 11.7 Hz), 6.58 (d, 2H, Hi, $J = 8.4$ Hz), 7.01 (d, 2H, Hj, $J = 8.0$ Hz). $^{13}$C NMR (CDCl$_3$, 126 MHz) $\delta$ (ppm): 13.8, 19.2, 20.4, 35.1, 52.4, 78.1, 113.8, 128.0, 130.1, 143.7. MS (ES+) m/z = 229.07 [MH$^+$], 199.24, 184.74, 118.00, 91.06.

**4i** $(E)$-$N$-(2-nitro-1-($p$-tolyl)ethyl)-4-(phenyldiazenyl)aniline. Dark-red thick oil. IR (neat NaCl, cm$^{-1}$): $\nu = 3397$, 3064, 3035, 2956, 2924, 2860, 1605, 1558, 1516, 1460, 1434, 1411, 1377, 1339, 1313, 1278, 1240, 1186, 1142, 1100, 1075, 1024, 967, 919, 862, 834, 767, 722, 687, 668, 640, 548, 532. $^1$H NMR (CDCl$_3$, 500 MHz) $\delta$ (ppm): 2.34 (s, 3H, Ha), 4.75 (d, Hb, 2H, $J = 6.6$ Hz), 4.84 (d, Hc, 1H, $J = 6.7$ Hz), 5.24 (m, Hd, 1H), 6.69 (d, 2H, He, $J = 8.9$ Hz), 7.20 (d, 2H, Hf, $J = 7.9$ Hz), 7.28 (d, 2H, Hg, $J = 8.1$ Hz), 7.39 (t, 1H, Hh, $J_o = 7.3$ Hz, $J_m = 1.3$ Hz), 7.46 (t, 2H, Hi, $J = 7.9$ Hz), 7.79 (d, 2H, Hj, $J_o = 8.9$ Hz, $J_m = 2.0$ Hz), 7.81 (d, 2H, Hk, $J = 8.3$ Hz). $^{13}$C NMR (CDCl$_3$, 126 MHz) $\delta$ (ppm): 21.1, 56.1, 79.9, 113.5, 122.4, 125.1, 126.3, 129.0, 129.9, 130.1, 133.9, 138.8, 145.7, 148.3, 153.0. MS (ES+) m/z = 361.18 [MH$^+$], 197.18, 118.04, 91.90, 77.00, 64.97.

**4j** 1-(4-((1-(4-methoxyphenyl)-2-nitropropyl)amino)phenyl)ethanone. Yellow-red thick oil. IR (neat NaCl, cm$^{-1}$): $\nu = 3348$, 3059, 3004, 2962, 2939, 2906, 2839, 1656, 1602, 1556, 1514, 1458, 1439, 1426, 1388, 1365, 1307, 1281, 1254, 1184, 1147, 1122, 1083, 1034, 960, 876, 833, 759, 669, 636, 597, 532. $^1$H NMR (CDCl$_3$, 500 MHz) $\delta$ (ppm): 1.56 (d, 3H, Ha, $J = 6.8$ Hz), 2.46 (s, 3H, Hb,), 3.79 (s, 3H, Hc,), 4.80 (t, 1H, Hd, $J = 7.9$ Hz), 4.99 (m, 1H, He), 5.05 (d, 1H, Hf, $J = 6.8$ Hz), 6.56 (d, 2H, Hg, $J = 8.9$ Hz), 6.89 (d, 2H, Hh, $J = 8.8$ Hz), 7.21 (d, 2H, Hi, $J = 8.7$ Hz), 7.74 (d, 2H, Hj, $J = 8.8$ Hz). $^{13}$C NMR (CDCl$_3$, 126 MHz) $\delta$ (ppm): 14.1, 26.1, 55.3, 59.7, 86.3, 112.8, 114.6, 128.0, 130.5, 150.1, 159.8, 196.4. $^1$H NMR (CDCl3, 500 MHz) $\delta$ (ppm): 1.52 (d, Ha, 3H, $J = 6.5$ Hz), 2.45 (s, Hb, 3H), 3.78 (s, Hc, 3H), 4.85 (t, 1H, Hd, $J = 7.8$ Hz), 4.94 (m, 1H, He), 5.05 (d, 1H, Hf, $J = 6.8$ Hz), 6.54 (d, 2H, Hg, $J = 8.9$ Hz), 6.88 (d, 2H, Hh, $J = 8.9$ Hz), 7.23 (d, 2H, Hi, $J = 8.6$ Hz), 7.76 (d,

2H, Hj, $J = 8.8$ Hz). $^{13}$C NMR (CDCl$_3$, 126 MHz) $\delta$ (ppm): 17.1, 26.0, 55.3, 60.3, 87.5, 112.7, 114.7, 128.0, 130.5, 150.1, 159.9, 196.4. MS (ES+) m/z = 329.10 [MH $^+$], 254.11, 148.03, 136.00, 117.01, 105.04.

**4k** 4-methyl-$N$-(2-nitro-1-(2-nitrophenyl)butyl)aniline. Red thick oil. IR (neat NaCl, cm$^{-1}$): $\nu = 3397$, 3054, 3025, 2978, 2936, 2920, 2876, 1621, 1593, 1555, 1523, 1488, 1456, 1405, 1377, 1332, 1329, 1300, 1269, 1256, 1186, 1173, 1142, 1110, 1075, 1014, 935, 884, 808, 760, 716, 633, 541. $^1$H NMR (CDCl$_3$, 500 MHz) $\delta$ (ppm): 0.94 (t, 3H, Ha, $J = 7.4$ Hz), 1.66 (m, 1H, Hb), 2.08 (m, 1H, Hc), 2.2 (s, 3H, Hd), 4.46 (broad, 1H, He), 4.63 (m, 1H, Hf), 4.73 (m, 1H, Hg), 6.45 (d, 2H, Hh, $J = 8.5$ Hz), 6.90 (d, 2H, Hi, $J = 8.7$ Hz), 7.18 (m, 2H, Hj), 7.46 (m, 2H, Hk). $^{13}$C NMR (CDCl$_3$, 126 MHz) $\delta$ (ppm): 10.4, 20.4, 24.8, 60.2, 94.6, 114.1, 122.4, 128.1, 128.6, 129.8, 132.3, 137.4, 143.2. $^1$H NMR* (CDCl3, 500 MHz) $\delta$ (ppm): 0.97 (t, 3H, Ha, $J = 7.3$ Hz), 1.90 (m, 1H, Hb), 2.14 (m, 1H, Hc), 4.24 (broad, 1H, Hd), 4.68 (m, 1H, He), 4.80 (m, 1H, Hf), 6.44 (d, 2H, Hg, $J = 8.5$ Hz), 6.92 (d, 2H, Hh, $J = 9.1$ Hz), 7.20 (m, 2H, Hi), 7.46 (m, 2H, Hj). $^{13}$C NMR (CDCl$_3$, 126 MHz) $\delta$ (ppm): 10.7, 22.4, 24.8, 60.4, 93.7, 114.2, 122.4, 128.4, 128.6, 129.8, 132.2, 137.0, 143.4. MS (ES+) m/z = 314.12, 274.11, 210.04, 178.06, 131.10, 108.09, 93.04, 91.03, 60.81.

## ACKNOWLEDGEMENTS

The authors thank the University of Malta for the equipment and Prof. Robert M. Borg for assistance with the acquisition of the NMR spectra.

### Funding

The authors received no funding for this work.

### Competing Interests

Giovanna Bosica is an Academic Editor for PeerJ.

### Author Contributions

- Giovanna Bosica conceived and designed the experiments, analyzed the data, contributed reagents/materials/analysis tools, authored or reviewed drafts of the paper, approved the final draft.
- Ramon Zammit performed the experiments, analyzed the data, prepared figures and/or tables.

### Data Availability

Copies of spectra used in the characterization of compounds are reproduced in the Supplemental Files.

### Supplemental Information

Supplemental information for this article can be found online at http://dx.doi.org/10.7717/peerj.5065#supplemental-information.

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
