# Peer review of "One-pot multicomponent nitro-Mannich reaction using a heterogeneous catalyst under solvent-free conditions"

_PeerJ, doi:10.7717/peerj.5065_

## Round 0.1 · original submission · Minor Revisions

The manuscript is acceptable for publication with minor revisions as suggested clearly by the reviewers' comments. The experiments were well planned, the results and analysis sound. The work presents an exciting and new, green, solvent-free method to prepare β-nitroamines and has potential to impact preparation of even more complex systems in this important class of molecules.

It would be instructive for the authors to include a scheme describing a generic Mannich reaction in the introduction. Particular attention to providing data assuring the reaction is completely heterogeneous through a method mentioned by the reviewer (hot filtration test) or one in the reference cited below. [1] The cited review focuses on homogeneous reactions but the tests for heterogeneous reactions are similar. In Equation 2 the abbreviation RMM has not been defined and this change would improve understanding for readers not familiar with the calculation. Also, are the solvents used in preparation of the catalyst loaded resin included in the calculations of E factor or atom economy? If not, this should be addressed and made clear in the text. Some of the substrates notations are incorrect, e.g. 2b-o (Table 5) resulted overall in short reaction times, …” However, Table 5 clearly shows that there are no products formed in the 2n and 2o reactions. In addition, please consider the reviewers’ comments with respect to specific changes that should be made to the manuscript. We look forward to receiving your revised manuscript.

1. Crabtree, R. H., “Resolving heterogeneity problems and impurity artifacts in operationally homogeneous transition metal catalysts.” Chemical reviews, 2011, 112(3), 1536-1554. DOI: 10.1021/cr2002905

·

Basic reporting

no comment

Experimental design

no comments

Validity of the findings

no comments

Additional comments

Manuscript id: Peer J -23415
Title: One-pot multicomponent nitro-Mannich reaction using a
heterogeneous catalyst under solvent-free conditions

Authors: Giovanna Bosica and Ramon Zammit

An environmentally-friendly, one-pot multicomponent reaction of various aldehydes, amines and nitroalkanes for the synthesis of β-nitroamines is described using Amberlyst A-21 supported CuI catalyst.


The Ms can be accepted for publication after minor revision.

1.One of the minor problems of this manuscript is English, which needs through editing.

2. Delete Fig 2. as advantages are summarized in conclusion.

·

Basic reporting

The article meets the standards for being published in PeerJ.

Experimental design

The article meets the standards for being published in PeerJ.

Validity of the findings

The article generally meets the standards for being published in PeerJ.

However, there is one experiment that should be carried out and reported to fully support the statement that the catalyst is truly heterogeneous. Indeed, the only way to acertain that the catalytic activity is strictly and solely the effect of the solid (and not for example from the 1% Cu that is leaching) is to perform a "hot filtration test". The reaction should be stopped at an early stage, when conversion has started but did not reach high values. Then the catalyst is removed (e.g. by filtration) and the filtrate is further allowed to react in the exact same conditions. If conversion value do not evolve at all, then the heterogeneous nature of the catalysis is confirmed.

When mechanistic insights are provided, it is not always clear what is speculation and what is known from the litterature (e.g. 153-155). In the latter case, a reference would be needed.

Additional comments

Though the paper is clearly of high quality (and interestingly includes valuable insights on the "greenness" of the process), I would like to make some suggestions.
1. abstract "catalysts can be reused up to 8 times". This statement is misleading. Authors have tested 8 recycling and found only minor deactivation (10%). But it is for the next user to decide if they want to recycle the cat further. Statement should be something linke "The catalyst has been tested up to 8 times with only a minor activity loss".
2. a genera Scheme of the Mannich reaction would nicely complement the first paragraph of the intro
3. Line 45-46. It is not correct to state that stoechimetric amount of catalyst is used. A catalyst is by definition not stoechiometric.
4. Line 54 "have substituted". What do authors mean? Industrially, everywhere, heterogeneous catalysts are used and not homogeneous cats? Only 1 specific paper is cited to support the sentence... Or maybe authors meant "have been shown to also catalyse"...
5. 68. Please clarify the sentence "which could be immobilized on a solid". Indeed, i the table, I see highly soluble species like KOH, Na2CO3, etc.
6. line 241. What does "nano-sized copper salt" means?

Reviewer 3 ·

Basic reporting

Comments for the authors

The authors describe an environmentally-friendly one pot multicomponent reaction of aldehydes, amines and nitroalkanes using the Amberlyst A-21 supported copper (I) iodide. The present nitro-Mannich type reaction tolerated versatile aldehydes and amines and regioselectively afforded the beta-nitroaminoalkanes in high yields. Furthermore, the reaction was achieved in a solvent-free condition. The authors also show the environmental acceptability of the nitro-Mannich reaction by calculating both the atom economy and the E-factor (as shown in eq 1 and 2). The catalyst seems to be practical and useful according to the investigation of Table 8 (Comparison of E-factore of the catalytic systems) and Table 9 (which shows the percentage of copper ions that leached into the solution from the heterogeneous catalyst). The manuscript seems to be suitable for the article of the PeerJ after minor revisions.

1) Text: line 33: synthesis via umpolung chemistry ----- synthesis of umpolung chemistry
2) Experimental Section: please check the 1H NMR.
4e: 1H NMR: 2H is missing.
4g: 1H NMR: 1H is too much.
4j: 1H NMR: 1H is too much.
4k: 1H NMR: 2H is missing.
3) The authors should cite the reference on the Aza-Henry reaction of nitriles with copper iodide/cesium carbonate/DBU in nitromethane (Org. Lett. 2018, 20, 1130).

Experimental design

good

Validity of the findings

good

---

## Round 0.2 · accepted · Accept

Thank you for addressing the comments and taking the suggestions and concerns of the reviewers in your revised manuscript. The work is an important contribution to the green catalysis literature.

#